# Repetitive Transcranial Magnetic Stimulation targeted with MRI based neuro-navigation in major depressive episode: a double-blind, multicenter randomized controlled trial

Bruno Millet[1]*, Ghina Harika-Germaneau[2], Redwan Maatoug[3], Florian Naudet [4,5], Jean-Michel Reymann[6], Valérie Turmel[6], Jean-Marie Batail[7], Jacques Soulabaille[7], Nematollah Jaafari[2], Dominique Drapier[7]

1 Institut du Cerveau et de la Moelle, UMR, CNRS, INSERM, Sorbonne Université et Département de Psychiatrie Adulte, Groupe Hospitalier Pitié-Salpêtrière, Paris, France, 2 CNRS, Université de Poitiers, Université de Tours, CeRCA, Poitiers, France; Unité de Recherche Clinique Pierre Deniker du Centre Hospitalier Henri Laborit, Poitiers, France, 3 Department of Psychiatry, Pitié-Salpêtrière Hospital, Public Hospitals of Sorbonne University, Paris, France, 4 Univ Rennes, CHU Rennes, Inserm, EHESP, Irset (Institut de recherche en santé, environnement et travail) - UMR_S, Rennes, France, 5 Institut Universitaire de France (IUF), Paris, France, 6 Univ Rennes, CHU Rennes, Centre d'investigation Clinique (CIC) Inserm, Rennes University Hospital, Rennes, France, 7 Adult Psychiatry Department, Guillaume-Régnier Hospital, Univ Rennes, CHU Rennes, Centre d'investigation Clinique (CIC) Inserm, Rennes, France

* b.millet@aphp.fr

## Abstract

### Context

High-frequency (HF) transcranial magnetic stimulation (rTMS) of the left dorsolateral prefrontal cortex (DLPFC) is widely used in Major Depressive Episode (MDE). Optimization of its efficacy with a neuro-navigation system has been proposed based on a small randomized controlled trial (RCT) supporting a large effect.

### Method

This evaluator- and patient-blind, multicenter RCT assessed the superiority in terms of efficacy of 10 HF rTMS sessions of the left DLPFC targeted with MRI based neuro-navigation versus similar sessions targeted by the standard 5 cm technique. The study was conducted between January 2013 and April 2017, at 4 hospitals centers in France where both in- and out- patients with MDE were included. Randomization was computer-generated (1:1), with allocation concealment implemented within the e-CRF. The main outcome measure was the percentage of responders 44 days (D44) after the rTMS session. Secondary outcomes were percentage of remitters, Beck Depression Inventory and psychomotor retardation assessed with Salpêtrière retardation rating scale (SRRS) for depression at D14 and D44. The results are presented along with their 95% confidence intervals.

**Data availability statement:** This study involves human participant data that cannot be publicly shared due to French regulatory constraints, as it includes potentially identifying or sensitive patient information. In the European context, under GDPR, it is not possible to openly share this sensitive data, even when it is pseudonymized. Furthermore, this is an old study where data sharing was not pre-planned in the documents approved by the IRB. Any data request will need to seek non-opposition from study participants, as practiced in the French context. Therefore, datasets will be made available upon request, subject to the non-opposition of study participants. This non-opposition process may incur minimal additional costs. Data requests can be sent to the Data Privacy Officer at Rennes University Hospital at DRI@chu-rennes.fr.

**Funding:** This study was funded by Syneika. The funders had no role in study design, data collection and analysis, decision to publish, or preparation of the manuscript.

**Competing interests:** All authors have completed the ICMJE uniform disclosure form available at http://www.icmje.org/coi_disclosure.pdf, which can be requested from the corresponding author. The authors declare that they have no conflicts of interest, except for Prof. Millet, who holds stocks in Syneika company. Dr Nauczyciel who is syneika CEO spouse worked on Prof. Millet's team and helped with data acquisition without fulfilling authorship criteria. These potential conflict of interest do not alter our adherence to PLOS ONE policies on sharing data and materials.

## Results

105 patients were randomized and 92 were evaluable with respectively 45 patients in the neuronavigation group and 47 in the standard group. A treatment response was observed for 14 (31.8%) of 44 patients analyzed in the intervention group, and for 16 (35.6%) of 45 patients analyzed in the control group with no statistical difference (relative risk 0.89; 95% confidence interval, [0.50;1.61]). No difference was evidenced for secondary outcomes at D44 whether it concerns remission at D44 (relative risk, 0.82; 95% CI, 0.36 to 1.88), or BDI results (difference in means, 0,01; 95% CI, -3.06 to 3.26), or SRRS results (difference in means, 0.11; 95% CI, -2.42 to 5.02). Similar results were observed at D14. Rates of adverse events were similar in both groups with 23 (47.9%) and 1 (2.1%) of adverse events and serious adverse events in the neuro-navigation group versus 20 (40.8%) and 0 (0%) in the standard group.

## Discussion

This study failed to reproduce previous findings supporting the use of neuro-navigation system to optimize rTMS efficacy. Limitations of this study includes a small sample size and a number of rTMS sessions that may appear substandard in 2025.

## Trial registration:

NCT01677078

## Introduction

Major depressive episodes (MDE) represent a major public health problem associated with increased functional disability and mortality. Limited efficacy of antidepressants [1] have triggered a specific interest in new therapeutics such as noninvasive neuromodulation, including repetitive transcranial magnetic stimulation (rTMS).

rTMS is the most widely used and studied form of TMS and uses an alternating current passed through a metal coil placed against the scalp to generate rapidly alternating magnetic fields, which pass through the skull nearly unimpeded and induce electric currents that depolarize neurons in a focal area of the surface cortex [2–4]. Although the mechanism of action is still debated, one hypothesis is that stimulation of discrete cortical regions alters pathologic activity within a network of grey matter brain regions that are involved in depression and connected to the targeted cortical sites. This is supported by functional imaging studies that show rTMS can change activity in brain regions distant from the site of stimulation [5]. Furthermore, rTMS has many molecular effects, including increased monoamine turnover and normalization of the hypothalamic pituitary axis [6]. In one neuroimaging study of depressed patients, a prefrontal serotonin deficiency at baseline normalized after treatment with rTMS [7].

The effect of rTMS appears to vary according to the frequency; high frequency (HF) stimulation is thought to excite the targeted neurons (and is typically used to

activate the left prefrontal cortex), whereas low frequency (LF) stimulation appears to inhibit cortical activity (and is usually directed at the right prefrontal cortex) [8]. Consistent with this hypothesis, a review examined 66 studies in depressed patients who were treated with TMS targeting the dorsal lateral prefrontal cortex and found that HF TMS generally increased regional cerebral blood flow and that LF TMS generally decreased regional cerebral blood flow [9].

Numerous meta-analyses of randomized trials have found evidence that rTMS provides a clinically relevant benefit to patients with treatment resistant depression (TRD) [4,10–12] such as Brunoni et al. who identified in a 2017 network meta-analysis that the rTMS was more effective than sham stimulation. However, most studies in the rTMS literature are underpowered, with a risk of excess significance bias [13,14]. As a consequence, the use of rTMS for MDD in clinical practice is still not officially recommended in several countries such as France. Conversely, in the United States of America, repetitive TMS has been approved by the Food and Drug Administration (FDA) to be used in a routine clinical setting for depression. In the United Kingdom, the National Institute for Health and Care Excellence (NICE) adopted a more nuanced position regarding its efficacy arguing that "the evidence on its efficacy in the short-term is adequate, although the clinical response is variable" [15]. Such divergent opinions about the evidence-based efficacy of this technique may in part reflect the numerous challenges in developing and validating such a complex stimulation technique which involves numerous parameters that can impact its efficacy and safety [16–18].

The accurate neuroanatomical location of the CPFDL is surely one of these parameters. While it is now considered as consensual across studies [19–21], it is challenging to specifically reach this region due to the interpersonal variability in terms of size and shape. Several methods have been developed recently to enable better targeting, ranging from the simplest methods (Standard method, 10–20 EEG localization) to increasingly sophisticated ones (neuronavigation with structural MRI, neuronavigation with functional MRI, Electric field modeling). Despite advances in targeting techniques, the superiority of one method over another remains, to date, unestablished, especially in routine clinical practice [19,20,22,23].

The standard method (1995) stipulates that for each patient the CPFDL region is over the left prefrontal cortex, determined by moving the TMS coil 5 cm anterior to the motor threshold location along a left superior oblique plane with a rotation point about the tip of the patient's nose [24]. Using a neuro-navigation system to perform a comprehensive quantification of target localization errors, Nauczyciel et al. [25] have identified 3 sources of error with this method: (i) the cap repositioning, (ii) the interexpert variability in coil positioning and (iii) the distance between the stimulated point and the expected target. Moreover a recent study [26] compare standard and neuronavigation targeting methodologies on distance, angle, and electric field magnitude values demonstrated that neuronavigation technology enables more precise and accurate TMS positioning, resulting in the intended stimulation intensities at the targeted cortical level. Despite this difference, the authors highlight the need for prospective studies to determine the clinical superiority of one method over the other.

To our knowledge, only one study has compared the two approaches in a randomized controlled manner. Fitzgerald et al. designed a randomized trial of repetitive TMS targeted with magnetic resonance imaging (MRI) based neuro-navigation in TRD. In this study including 51 patients with treatment resistant depression the neuro-navigational approach targeting a specific site at the junction of Brodmann areas 46 and 9 in the DLPFC based on each individual subject's MRI scan outperformed the standard localization group at 4 weeks [19].

We therefore designed this study to reproduce the findings from Fitzgerald et al. [19] in the French context using a larger sample size and an in-house developed French neuro-navigation system (Syneika; syneika.fr) which takes its originality in the automatic location of the left DLPFC. Thus, our definition of reproducibility includes both the concept of result reproducibility and generalizability, as outlined by Goodman et al. [27]. A first objective of this multicenter randomized controlled trial was to demonstrate the superiority in terms of efficacy over the depressive symptoms of HR-rTMS of the left DLPFC targeted with MRI based neuro-navigation versus the standard localization technique in MDD using the MADRS score.

## Methods

### Design

We conducted a multicentric (four hospitals in France (Rennes, Paris, Poitiers, St Avé), double-blind, randomized (1:1) controlled trial. Patients and raters were blinded. Patients were randomized to receive either neuronavigated or standard method location. Both groups received two weeks of rTMS. The protocol (see supporting information for the French version and an English summary) was approved by the Human Investigation Committee of Brest, France, on March 20, 2012 (2011-A01272-39). The study was registered on ClinicalTrials.gov (NCT01677078) on August 31, 2012.

### Eligibility criteria for participants

All patients were in- and out- patients, aged from 18 to 65, right-handed, suffering major depressive episode according to DSM-IV criteria. At screening, eligible participants had a score of 21 or more on the Montgomery and Åsberg Depression Rating Scale (MADRS). Their antidepressant treatment should not have been modified for at least 3 weeks. Benzodiazepines had to be stopped or possibly replaced by other anxiolytics. Mood stabilizers, when previously prescribed, were not stopped.

Non-inclusion criteria were: 1/ depression with psychotic features, 2/ co-occurrence of schizophrenia, alcohol or other substance abuse or dependence, 3/ stage 5 treatment resistant depression according to Thase and Rush, 4/ involuntary hospitalization, 5/ patients under custodianship, 6/ contraindication to MRI or rTMS (i.e., epilepsy, neurological disorder, metallic prosthetic material within the brain, pacemaker and intra-ocular ferromagnetic material), 7/ pregnancy. Patients with high suicidal risk (item 10 MADRS> 3) were included only if they were in-patients.

Concerning concomitant medications, antidepressant regimen was not modified unless therapeutic emergency and any changes during the protocol had to be reported in the observation notebook.

In case of major suicidal risk, patients had to be hospitalized in accordance with good clinical practice.

All patients gave their written informed consent.

### Interventions for each group

All included patients were randomized before receiving a 3DT1 MRI.

### rTMS parameters

The first rTMS session, defining day 1 of the protocol (D1) had to be completed no later than 15 days after randomization. Patients were treated every weekday from Monday to Friday with no treatment on weekends, amounting to 10 sessions of HF-rTMS delivered under the supervision of an investigator aware of treatment arm.

The rTMS protocol used a frequency of 20 Hz with 40 pulses per train, 80 trains, and an inter-train interval of 10 seconds, resulting in a total of 3,200 pulses per session. The treatment intensity was set at 110% of the patient's resting motor threshold (RMT) as determined by visual observation. The rationale for using 20 Hz was based on data from Speer et al. (2000) [28], which demonstrated that two weeks of daily 20-Hz rTMS over the left prefrontal cortex at 100% of the motor threshold resulted in a sustained increase in regional cerebral blood flow in the bilateral frontal, limbic, and paralimbic regions implicated in depression. This decision was also supported by multiple studies on the efficacy of 20-Hz rTMS administered over the left prefrontal cortex [29–31].

### 5 cm standard method

The standardized treatment location followed the 5 cm rule over the left prefrontal cortex. For this procedure, a blank head cap was positioned in a tight-fitting manner. Once the motor hotspot and RMT are identified, the motor cortex serve as a reference point, The target was determined by moving the TMS coil 5 cm anterior to the MT location along a left superior

oblique plane with a rotation point about the tip of the patient's nose [24]. A reliable cap position was ensured by placing it according to the distance between the nasion and the beginning of the cap. We called this study group Standard method.

### MRI-based neuro-navigation

According to Rajkowska and Godman-Rakic [32], the left DLPFC localization on each patient's MRI scan is defined as a spot between the center of Broadman area (BA) 9 and the border of BA 9 and 46, over the junction of the middle and anterior thirds of the middle frontal gyrus.

The Syneika device's localization method implies a template brain scan (named Colin27) on which the DLPFC has been localized by a neuroanatomy expert. Patient's brain, automatically segmented from MRI scan, is then registered to Colin27 template thanks to rigid and non-rigid transforms. These transforms are finally used to report the targeted spot identified on Colin27 template scan to the patient's MRI space.

After the patient's head is registered with MRI space, co-registration accuracy is tested by placing the tracker (Syneika's device integrates a MicronTracker device by ClaroNav) over known anatomical landmarks.

### Pre-specified primary and secondary outcome measures

The pre-specified primary endpoint with respect to efficacy was the response rate, defined by a 50% improvement from day 1 (D1) to day 44 (D44), as measured by the MADRS. Any change in a patient's usual antidepressant treatment during the study was considered as failure on the primary and the secondary outcomes. The pre-specified secondary endpoints were 1/ the response rate at day 14, 2/ remission rate (i.e., patients with a MADRS score of 8 or less) at D14 and D44, 3/ Beck Depression Inventory (BDI) at D14 and D44, and 4/ psychomotor retardation assessed with Retardation Rating Scale for Depression (ERD) at D14 and D44. There were no changes in trial outcome registration after the trial has started.

### Sample size calculations

The expected rate of responders in the standard rTMS arm was 35%. We hypothesized that neuro-navigation-guided rTMS may increase this proportion up to 70% based on the large effect size observed in the princeps study [19]. Therefore, 51 evaluable patients per arm were necessary to guarantee a 95% power and a 0.05 Type-I error (two-sided). Considering that 15% of patients may be lost to follow-up, the total number of subjects needed was then of 120 (60 per group).

### Randomization and mechanism used to implement the random allocation sequence

For allocation of the participants, a computer-generated list of random numbers was used. Randomization sequence was created using SAS statistical software and was stratified by center with a 1:1 ratio.

The allocation sequence was concealed from the investigator in charge of including and evaluating patients: information on treatment arm was electronically available only for the investigator in charge of the rTMS sessions, on a dedicated e-CRF page that the investigator in charge of evaluation could not access. This information was available only after the enrolled participants completed all baseline assessments and it was time to allocate the intervention.

### Blinding

Only investigators in charge of the rTMS sessions were aware of the allocated arm. Care providers, medical staff and outcome assessors were kept blinded to the allocation. When analyzing the data, statisticians were not blinded. Patients were also kept blind to the allocation. All patients wore a head cap with a mark representing the target identified using standard method. In the standard "arm" the coil was placed on this point, and the neuro-navigator screen showed the

position of the coil on the patient's brain and no target was shown on the neuro-navigator screen. In the "neuro-navigation" arm, the target on the cap was disregarded and the coil placed on the target indicated by the neuro-navigator screen.

### Statistical methods used to compare groups

The primary analysis was a modified intention-to-treat analysis (mITT) where all evaluable patients (with at least a D1 and D14 assessment) were included. Continuous variables are presented as means and standard deviations. Categorical variables are presented as the number of patients in each category and the corresponding percentages. Missing data at D44 were replaced using the Last Observation Carried Forward approach (this choice, not pre-specified in the protocol, was made before drafting the statistical analysis plan). The primary outcomes as well as all other categorical efficacy outcomes were compared using Chi2 (relative risks, absolute risk differences and 95% confidence intervals were reported). Safety outcomes were analyzed using Fisher's exact test. Continuous variables were compared using t-tests (mean differences and 95% confidence intervals were reported). Significance level was set out at 0.05. The primary analysis was completed by a sensitivity analysis on the per-protocol population defined as those who received the study treatment according to the protocol and who benefited from automatic localization for neuronavigated patients. Database was locked on November 2018, statistical analysis plan was drafted on November 2018, and the statistical analysis (attached as a supporting information) was completed on December 2018. All analyses were conducted with SAS statistical software, version 9.4 (SAS Institute). Reporting followed CONSORT 2010 (see checklist in supporting information).

### Important changes to methods after trial commencement, with reasons

As delay between randomization and rTMS sessions were often longer than the 15 weeks anticipated in the protocol, due to delay to obtain MRIs and delays to schedule rTMS, the protocol allowed inclusion of patients who already had an MRI done prior to study entry. Due to recruitment difficulties, inclusions were stopped by the sponsor (Rennes University Hospital) after 105 patients were included.

## Results

### Participant flow

105 participants were recruited starting in January 2013 (21/01/2013). Last visit of the last patient was on April 2017 (13/04/2017).

The mITT population comprised 92 patients (88% of the randomized patients with 45 (87%) in the neuro-navigation group and 47 (89%) in the standard group). The per-protocol population comprised 74 patients (70% of the randomized patients with 32 (62%) in the neuro-navigation group and 42 (79%) in the standard group). Fig 1 summarizes the participant flow in this study. Table 1 displays initial characteristics of both groups.

### Efficacy results

**Primary outcome.** No difference was evidenced (p = 0.58) in terms of treatment response with 14 (31.8%) responders in the intervention group, and 16 (35.6%) in the control group (relative risk, 0.89; 95% CI, 0.50 to 1.61 and absolute risk difference, 3.7%; 95% CI, -0.16 to 0.23]. Similar results were observed in the per-protocol population (Table 2).

**Secondary outcomes.** No difference was evidenced for response at both D14 and D44 as displayed in Table 2. Difference in means was of 0,01; 95% CI, -3.06 to 3.26 for BDI at D44 and of 0.11; 95% CI, -2.42 to 5.02 for ERD at D44. Fig 2 displays results observed on MADRS (Fig 2A), BDI (Fig 2B) at each time point.

**Safety results.** Rates of adverse events were similar in both groups with 23 (44%) and 1 (2%) of adverse events and serious adverse events in the neuronavigation group versus 20 (38%) and 0 (0%) in the standard group (p-values of 0.55 and 0.49 respectively). Table 3 presents all adverse events.

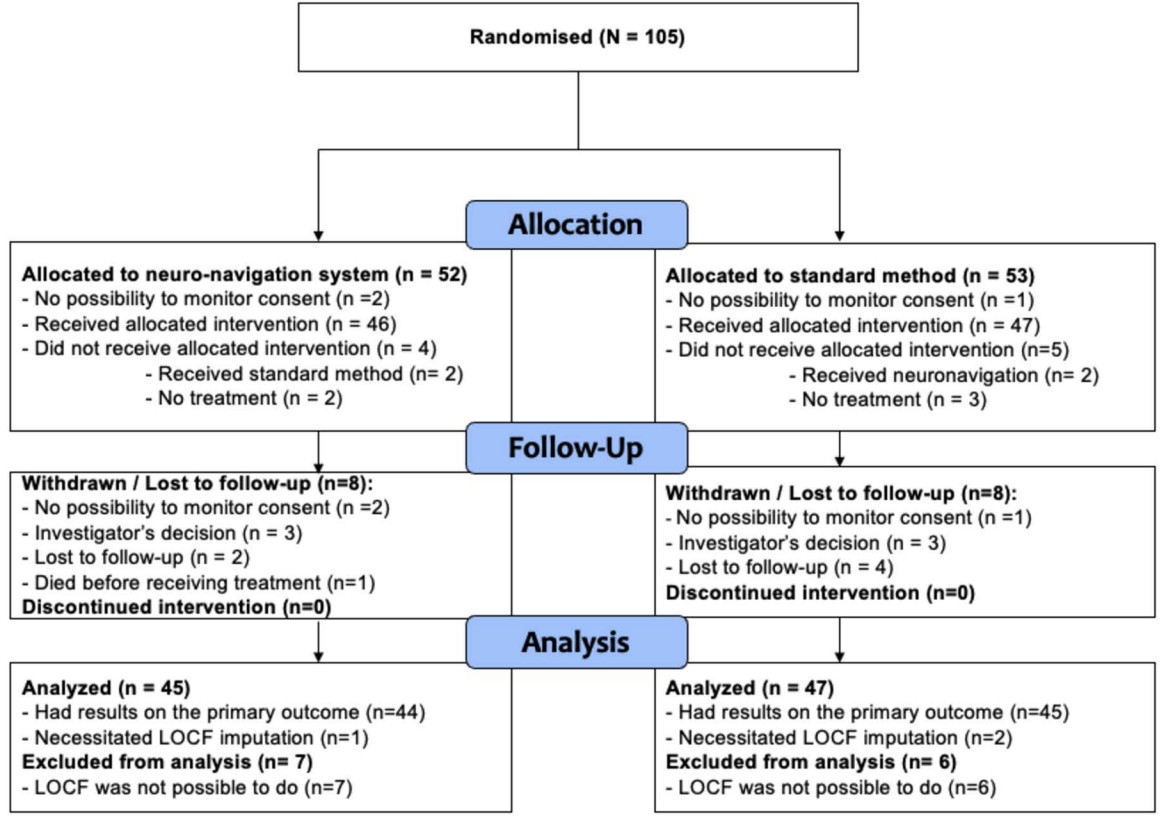

**Fig 1. Study flow chart.**

## Discussion

Our study failed to reproduce Fitzgerald et al.'s study result supporting the use of neuro-navigation system to optimize rTMS efficacy [19]. Several reasons could explain our inability to reproduce such result. First, Fitzgerald et al.'s study was performed on a single clinical center and we do not have information on the number of clinicians that administered rTMS treatment and their expertise. Our study is subject to greater variability in rTMS operators due to its multicentric design, involving multiple clinicians at each center and reflecting real clinical conditions with differences in equipment and timing. In their 2022 study, Caulfield et al. [26] compared neuronavigation with the standard placement method, assessing the influence of TMS operators on elastic cap placement during each stimulation day. They examined its effects on the distance, roll/pitch angle, and yaw angle deviations from the DLPFC target. The results revealed significant differences in cap placement measurements between operators using the standard method, but no such differences were observed with neuronavigation. Neuronavigation demonstrated consistent accuracy, unaffected by operator variability. Overall, these findings emphasize the operator's influence on the standard method, which may contribute to variability in our study.

Second, Fitzgerald's study included a relatively small sample size, with 24 patients in the neuro-navigational group and 27 in the standard group. While our study involved a larger sample, it remains relatively small, as it was designed to detect a somewhat large effect of neuro-navigated rTMS. A lack of statistical power may therefore explain our negative results. In addition, the modified ITT analyses primarily reflect the implementation of the neuro-navigation system in our study and the constraints encountered during its execution. Notably, we faced several deviations, including significant delays between randomization and Day 1 due to technical and logistical challenges. These delays may have led to spontaneous

**Table 1.** Demographic and baseline clinical variables (n = 102).

| | Whole sample (n = 102˚) | Neuro-navigation group (n = 50˚) | Standard group (n = 52˚) |
|---|---|---|---|
| Demographic | | | |
| Age (years) | 50.3 ± 10.9 | 50.3 ± 11.1 | 50.3 ± 10.8 |
| Sex (men) | 40 (39.2%) | 20 (40.0%) | 20 (38.5%) |
| Family situation | | | |
| Single | 31 (30.4%) | 19 (38%) | 12 (23.1%) |
| Couple | 62 (60.8%) | 25 (50.0%) | 37 (71.2%) |
| Separated | 8 (7.8%) | 6 (12%) | 2 (3.8%) |
| Vidower/ Widow | 1 (1%) | 0 (0%) | 1 (1.9%) |
| One child or more | 70 (68.6%) | 35 (70%) | 35 (67.3%) |
| Degree of education | | | |
| Middle school | 3 (3%) | 3 (6%) | 0 (0%) |
| High school | 27 (26.7%) | 14 (28%) | 13 (25.5%) |
| Bachelor | 16 (15.8%) | 6 (12%) | 10 (19.6%) |
| Master | 8 (7.9%) | 5 (10%) | 3 (5.9%) |
| PhD | 5 (5%) | 3 (6%) | 2 (3.9%) |
| Professional activity | | | |
| Work | 26 (25.7%) | 12 (24.5%) | 14 (26.9%) |
| Unemployed | 12 (11.9%) | 4 (8.1%) | 8 (15.4%) |
| Retired | 18 (17.8%) | 8 (16.3%) | 10 (19.2%) |
| Sick leave | 45 (44.6%) | 25 (51%) | 20 (38.5%) |
| Medical history | | | |
| Duration mood episode weeks) | 16.4 | 17.5 | 15.3 |
| MADRS score at baseline | 28.0 | 28.6 | 27.5 |
| MADRS score item 10 | 1.7 | 1.8 | 1.6 |
| Thase et Rush staging | | | |
| Stage 1 | 12 (12%) | 6 (12.2%) | 6 (11.8%) |
| Stage 2 | 45 (45%) | 23 (46.9%) | 22 (43.1%) |
| Stage 3 | 35 (35%) | 17 (34.7%) | 18 (35.3%) |
| Stage 4 | 6 (6%) | 2 (4.1%) | 4 (7.8%) |
| Stage 5 | 2 (2%) | 1 (2%) | 1 (2%) |
| Mood disorder type | | | |
| Single-episode major depressive disorder | 23 (22.5%) | 14 (28%) | 9 (17.3%) |
| Recurrent major depressive disorder | 59 (57.8%) | 26 (52%) | 33 (63.5%) |
| Bipolar disorder | 20 (19.6%) | 10 (20%) | 10 (19.2%) |
| Year of first mood episode | 2000 | 1997 | 2002 |
| Time between first mood episode and inclusion in the study (years) | 16 | 19 | 14 |
| Type of first mood episode | | | |
| Depressive | 99 (98%) | 48 (98%) | 51 (98,1%) |
| Hypomanic | 1 (1%) | 0 (0%) | 1 (1.9%) |
| Manic | 1 (1%) | 1 (2%) | 0 (0%) |
| Number of previous depressive episodes | | | |
| 0 | 9 (8.9%) | 7 (14.3%) | 2 (3.8%) |
| 1 | 17 (16.8%) | 8 (16.3%) | 9 (17.3%) |

*(Continued)*

**Table 1.** (Continued)

| | Whole sample (n = 102*) | Neuro-navigation group (n = 50*) | Standard group (n = 52*) |
|---|---|---|---|
| 2–5 | 51 (50.1%) | 22 (44.9%) | 29 (55.8%) |
| > 5 | 24 (23.8%) | 12 (24.5%) | 12 (23.1%) |
| Number of previous hypo or manic episodes | | | |
| 0 | 82 (82.8%) | 39 (83%) | 43 (82.7%) |
| 1 | 6 (6.1%) | 3 (6.4%) | 3 (5.8%) |
| 2–5 | 10 (10.1%) | 5 (10.6%) | 5 (9.6%)1 |
| > 5 | 1 (1%) | 0 (0%) | 1 (1.9%) |
| History of suicide attempt | 52 (51%) | 28 (56%) | 24 (46.2%) |
| History of anxiety disorder | 39 (39%) | 20 (40%) | 19 (38%) |
| rTMS treatment in the past | 11 (10.8%) | 3 (6.0%) | 8 (15.4%) |
| Number of rTMS sessions in the past | | | |
| 1 | 6 (54.5%) | 0 (0%) | 6 (75%) |
| 2 | 2 (18.2%) | 2 (66.7%) | 0 (0%) |
| 5 | 2 (18.2%) | 0 (0%) | 2 (25%) |
| 6 | 1 (9.1%) | 1 (33.3%) | 0 (0%) |
| First-degree family history of mood disorder | | | |
| 0 | 38 (40%) | 18 (39.1%) | 20 (40.8%) |
| 1 | 47 (49.5%) | 23 (50%) | 24 (49%) |
| > 1 | 10 (10.5%) | 5 (10.9%) | 5 (10.2%) |
| First-degree family history of suicide | | | |
| 0 | 88 (92.6%) | 45 (95.7%) | 43 (89.6%) |
| 1 | 5 (5.3%) | 2 (4.3%) | 3 (6.3%) |
| > 1 | 2 (2.1%) | 0 (0%) | 2 (4.2%) |

Continuous variables are presented as means and standard deviations. Categorical variables are presented as the number of patients in each category and the corresponding percentages.

*Information of 3 randomized patients (2 in the neuro-navigation group and 1 in the standard group) could not be reported as their consent form could not be retrieved during study monitoring

improvement in some patients before the first session, potentially resulting in a less severe patient population compared to other studies. Despite these factors, the per-protocol analyses did not demonstrate the superiority of neuro-navigated rTMS. Importantly, the sample size in this study population remained larger than that of Fitzgerald's trial, which reported a very large effect size with 10/24 patients (42%) in the targeted group and 5/27 (18%) in the standard group met the response criteria at 4 weeks.

Third, the stimulation parameters were noticeably different in both studies. The rTMS treatment was applied at 10Hz with an intensity of 100% of resting motor threshold for a total of 20 sessions per patient in Fitzgerald et al. In comparison, in our study the rTMS treatment was applied at 20Hz with an intensity of 110% of resting motor threshold for a total of 10 sessions per patient. In particular, the limited number of sessions conducted in our study may have influenced the operator-dependent effects in standard positioning and it perhaps prevented a more pronounced differentiation between groups, as observed in the study by Fitzgerald et al. Indeed, previous research [33,34] supports the idea that a greater number of sessions can enhance overall efficacy.

Forth, the differences in neuro-navigation system could also explain why we found no difference between the two groups in our study. The localization method of the DLPFC used in Fitzgerald study is based on warping scan into

**Table 2. Response and remission criteria at D14 and D44.**

| | Neuro-navigation group | Standard group | p-value | RR [95%CI] | RD [95%CI] |
|---|---|---|---|---|---|
| **Modified Intention to Treat** | n = 45 | n = 47 | | | |
| D14 | | | | | |
| Responders | 14 (31%) | 15 (33%) | 0.88 | 0.97 [0.53; 1.78] | 0.01 [-0.18; 0.20] |
| Remitters | 7 (16%) | 12 (26%) | 0.24 | 0.61 [0.26; 1.41] | 0.10 [-0.06; 0.26] |
| D44 | | | | | |
| Responders | 14 (32%) | 16 (36%) | 0.71 | 0.91 [0.51; 1.65] | 0.03 [-0.16; 0.22] |
| Remitters | 8 (18%) | 10 (22%) | 0.63 | 0.84 [0.36; 1.93] | 0.03 [-0.13; 0.20] |
| **Per-protocol** | n = 32 | n = 42 | | | |
| D14 | | | | | |
| Responders | 9 (28%) | 13 (32%) | 0.74 | 0.91 [0.44; 1.86] | 0.03 [-0.18; 0.24] |
| Remitters | 4 (12%) | 10 (24%) | 0.22 | 0.53 [0.18; 1.52] | 0.11 [-0.06; 0.29] |
| D44 | | | | | |
| Responders | 10 (31%) | 15 (36%) | 0.69 | 0.88 [0.45; 1.68] | 0.04 [-0.17; 0.26] |
| Remitters | 5 (16%) | 9 (22%) | 0.50 | 0.73 [0.27; 1.97] | 0.06 [-0.12; 0.3] |

Variables are presented as the number of patients in each category and the corresponding percentages. RR = relative risk, RD = risk difference

Talairach space (by using SPM99), and then reporting target spot already defined in this space (-45;45;35) into the warped scan. In our study, the Syneika device's localization method implies a template brain scan (named Colin27) on which the targeted spot (DLPFC) has been localized by a neuroanatomy expert. Patient's brain, automatically segmented from MRI scan, is then registered to Colin27 template thanks to rigid and non-rigid transforms. These transforms are finally used to report the targeted spot identified on Colin27 template scan to the patient's MRI space. Using our method, we have estimated the error of this method at 12mm root mean square compared to the same localization manually done on each patient's MRI by a neuroanatomy expert, while no error estimation was calculated in Fitzgerald study. Furthermore, in Fitzgerald study, the patient's head is placed in a frame to prevent movements during the treatment, while in our study, no such movement preventing solution was used; the patient was only seated in a chair on which he was able to release his head. Our device was nevertheless able to detect large and lasting movements.

A final point that may explain the difference lies in the clinical characteristics of the patients. In our study, we included patients with more heterogeneous types of depression than in Fitzgerald's study. Twenty percent of our patients had bipolar depression, which was an exclusion criterion in the Fitzgerald study. A recent meta-analysis [33] suggests that high-frequency stimulation of the left DLPFC may not be optimal for bipolar disorder, potentially impacting overall efficacy. Despite these potential differences, we chose not to conduct subgroup analyses, as they carry a risk of alpha risk inflation and cannot be reliably interpreted when not planned a priori and/or when the results are negative, as in our case. The type of depression may have a different cerebral impact, affecting brain structure, function, and connectivity, which may influence target selection and response to treatments like rTMS [11,35]. It is important to note that the 5 cm rule does not always target the classic DLPFC areas (BA 9 and 46) [36]; instead, it reaches a transitional region of the DLPFC connected to the Default mode network and motor cortex. This could explain the different, and sometimes beneficial, responses to rTMS in a heterogeneous clinical population.

It is important to note that our result is, however, consistent with other studies that compared neuronavigation versus the other method. Hebel et al., 2021 [22] conducted a single-center, two-arm, randomized, double blind study with 37

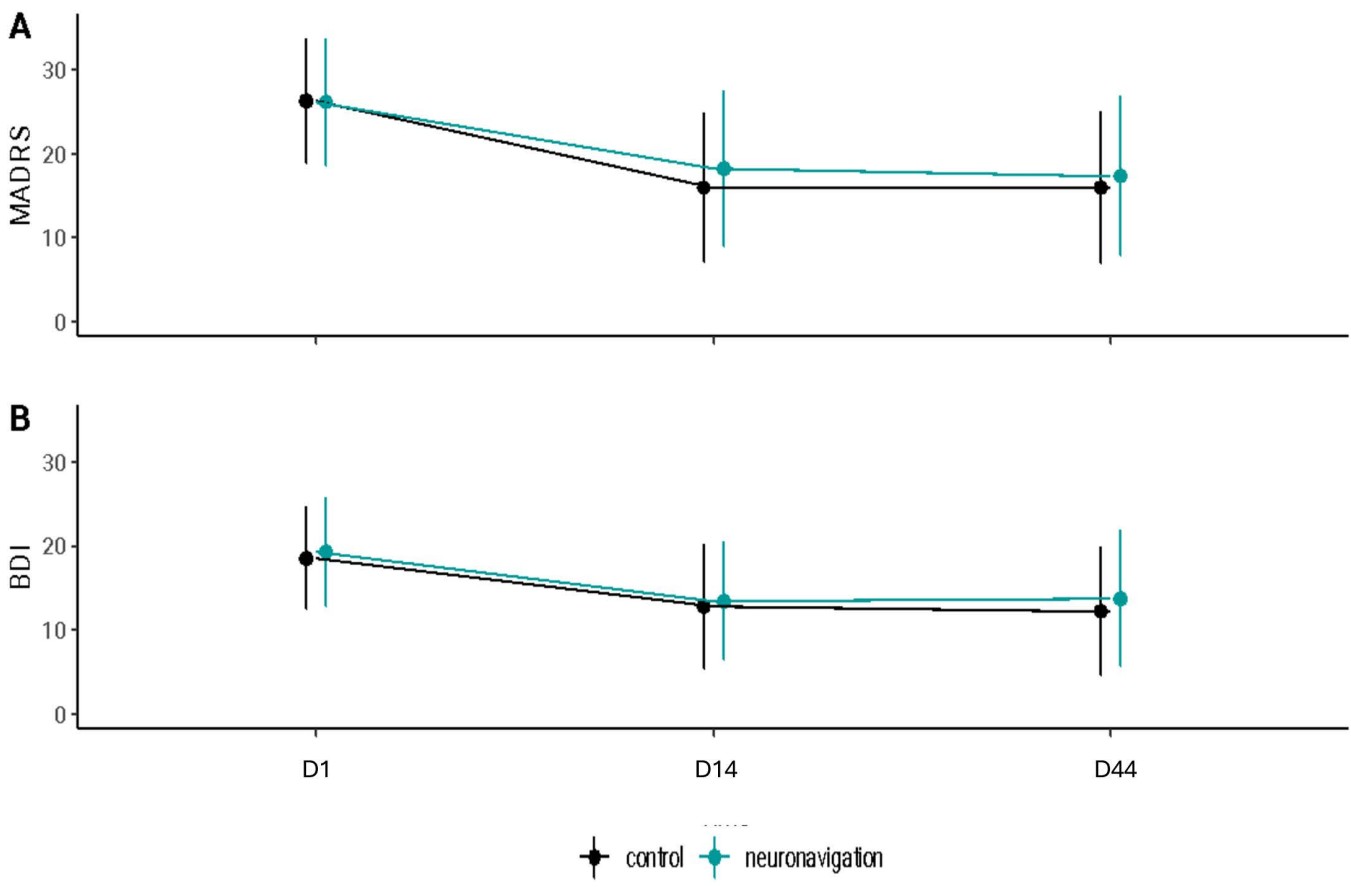

**Fig 2. Mean scores ( +/- Standard deviations) at baseline and at each visit (D1, D14, D44) in (a) Montgomery and Åsberg Depression Rating Scale, (b) Beck Depression Inventory and (c) Retardation Rating Scale for Depression scale.**

inpatients assigned to receive either neuronavigated iTBS or 10–20 EEG system-based F3 guided iTBS over the left dorsolateral prefrontal cortex (lDLPFC). The study did not demonstrate clinical superiority of neuronavigated localization compared to the 10–20 EEG system-based F3 approach. In another hand Tsukuda et al., 2024 [23] examines the effectiveness of three targeting methods for rTMS in treating depression among Japanese patients: the standard 5 cm rule, the F3 method, and neuronavigation. The study compares these approaches to determine their clinical efficacy and accuracy in targeting the lDLPFC in 16 depressive patients. Their results indicate that neuronavigation offers higher targeting precision, potentially leading to more consistent treatment outcomes, though differences in clinical effectiveness between methods were limited.

In conclusion, this study failed to identify any difference of efficacy of rTMS for depression when comparing neuronavigation-based targeting with the standard method. While neuronavigation theoretically offers more precise targeting of the DLPFC, no operator dependence, less target deviation [26], and a better distribution of the eclectic fields [37] this did not translate into superior clinical outcomes in our sample. Targeting methods have evolved considerably over time, enabling a personalized approach to depression treatment, particularly through functional MRI neuronavigation and electric field modeling. Despite the appeal and high precision of these approaches, their use in routine clinical practice remains limited due to cost and accessibility. As a result, establishing the most efficient localization method remains a clinical challenge.

**Table 3. Safety results.**

| | Neuro-navigation group | | Standard group | |
|---|---|---|---|---|
| | **Number of events** | **Number of patients** | **Number of events** | **Number of patients** |
| **Adverse events** | 42 | 23 (46%) | 23 | 20 (38%) |
| Eye Pain | 1 | 1 (2%) | 0 | 0 (0%) |
| Application site Pain | 4 | 4 (8%) | 3 | 3 (6%) |
| Device Intolerance | 0 | 0 (0%) | 1 | 1 (2%) |
| Medical Device site warmth | 1 | 1 (2%) | 1 | 1 (2%) |
| Feeling hot | 1 | 1 (2%) | 0 | 0 (0%) |
| Discomfort | 1 | 1 (2%) | 0 | 0 (0%) |
| Asthenia | 3 | 3 (6%) | 4 | 3 (6%) |
| Pain | 7 | 6 (12%) | 1 | 1 (2%) |
| Procedural headache | 1 | 1 (2%) | 0 | 0 (0%) |
| Headache | 13 | 11 (22%) | 11 | 11 (21%) |
| Amnesia | 1 | 1 (2%) | 0 | 0 (0%) |
| Cognitive disorder | 1 | 1 (2%) | 0 | 0 (0%) |
| Syncope | 1 | 1 (2%) | 0 | 0 (0%) |
| Dizziness | 1 | 1 (2%) | 0 | 0 (0%) |
| Presyncope | 2 | 2 (4%) | 0 | 0 (0%) |
| Muscle contractions involuntary | 1 | 1 (2%) | 0 | 0 (0%) |
| Anxiety | 0 | 0 (0%) | 1 | 1 (2%) |
| Hypomania | 0 | 0 (0%) | 1 | 1 (2%) |
| Mania | 2 | 2 (4%) | 0 | 0 (0%) |
| Sleep disorder | 1 | 1 (2%) | 0 | 0 (0%) |
| **Serious adverse events** | 1 | 1 (2%) | 0 | 0 (0%) |
| Intentional overdose | 1 | 1 (2%) | 0 | 0 (0%) |

The contradictory results between Fitzgerald et al. [20] and our study, as well as the limitation of both trials, should encourage the design of larger studies with optimal stimulation parameters and diverse patient populations. It would help clarify if specific subgroups could benefit more from one targeting method over the other.

## Supporting information

**S1 File. Protocol (French).**
(DOC)

**S2 File. Protocol (English translation).**
(DOCX)

**S3 File. Statistical analysis plan (French).**
(PDF)

## Author contributions

**Conceptualization:** Bruno Millet, Jean-Michel Reymann.

**Data curation:** Valérie Turmel.

**Formal analysis:** Florian Naudet, Valérie Turmel.

**Funding acquisition:** Bruno Millet.

**Investigation:** Bruno Millet, Ghina Harika-Germaneau, Redwan Maatoug, Jean-Marie Batail, Jacques Soulabaille, Nematollah Jaafari, Dominique Drapier.

**Methodology:** Florian Naudet, Jean-Michel Reymann, Valérie Turmel.

**Project administration:** Bruno Millet.

**Resources:** Valérie Turmel.

**Software:** Valérie Turmel.

**Supervision:** Bruno Millet.

**Validation:** Bruno Millet, Valérie Turmel.

**Visualization:** Redwan Maatoug, Valérie Turmel.

**Writing – original draft:** Bruno Millet, Redwan Maatoug, Florian Naudet.

**Writing – review & editing:** Bruno Millet, Ghina Harika-Germaneau, Redwan Maatoug, Florian Naudet, Jean-Michel Reymann, Valérie Turmel, Jean-Marie Batail, Jacques Soulabaille, Nematollah Jaafari, Dominique Drapier.

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
