## [Decision Letter · Decision Letter 0]

28 Jan 2025

PONE-D-24-60061

Repetitive Transcranial Magnetic Stimulation targeted with MRI based neuro-navigation in major depressive disorder: a double-blind, multicenter randomized controlled trial

PLOS ONE

Dear Dr. Naudet,

Thank you for submitting your manuscript to PLOS ONE. After careful consideration, we have decided that your manuscript does not meet our criteria for publication and must therefore be rejected.

I am sorry that we cannot be more positive on this occasion, but hope that you appreciate the reasons for this decision.

Kind regards,

Mu-Hong Chen, M.D., Ph.D.

Academic Editor

PLOS ONE

Reviewers' comments:

Reviewer's Responses to Questions

**Comments to the Author**

1. Is the manuscript technically sound, and do the data support the conclusions?

Reviewer #1: Yes

Reviewer #2: Partly

2. Has the statistical analysis been performed appropriately and rigorously? 

Reviewer #1: Yes

Reviewer #2: Yes

3. Have the authors made all data underlying the findings in their manuscript fully available?

Reviewer #1: Yes

Reviewer #2: Yes

4. Is the manuscript presented in an intelligible fashion and written in standard English?

Reviewer #1: Yes

Reviewer #2: Yes

5. Review Comments to the Author

Reviewer #1: This multicenter, double-blind randomized controlled trial compared MRI-based neuronavigation with the standard 5 cm method for rTMS targeting in major depressive disorder. While the study findings did not support the authors’ hypothesis, the results remain valuable in advancing our understanding of rTMS targeting. To enhance the quality of this manuscript, several major and minor concerns should be addressed.

Major Concerns

- The title refers to "major depressive disorder," and the recruitment criteria cite " major depressive disorder according to DSM-IV criteria in a current depressive episode." However, Table 1 indicates that approximately 60% of participants were diagnosed with persistent depressive disorder and 20% with bipolar disorder. Since persistent depressive disorder and bipolar disorder are distinct from major depressive disorder, the title should be revised accordingly.

- The term "persistent depressive disorder" is not part of DSM-IV terminology, which uses "dysthymic disorder." The authors should clarify whether they used DSM-IV or DSM-5 criteria.

- If participants with persistent depressive disorder were indeed diagnosed solely with dysthymic disorder, this implies that these individuals may not have experienced a major depressive episode. While the recruitment threshold of ≥21 on the Montgomery-Åsberg Depression Rating Scale appears appropriate, the diagnostic criteria and severity thresholds need to align clearly.

Minor Concerns

Abstract

- The authors report relative risks with 95% confidence intervals (CIs) but do not describe the statistical methods in the Methods section.

- Adverse event rates are described as "similar," yet no statistical results are reported for this comparison. Statistical tests should be included for consistency.

Introduction

- The authors aim to "reproduce the findings from Fitzgerald et al." However, their rTMS protocol (20 Hz with 40 pulses per train, 80 trains, inter-train interval of 10 seconds, 110% resting motor threshold, and 10 sessions) differs significantly from that used in Fitzgerald et al. These differences should be explained and acknowledged in the Introduction.

Methods

- The rationale for using 20 Hz rather than the more common 10 Hz protocol should be provided. While 20 Hz is not inherently problematic, justification for its selection is necessary.

- Details on statistical methods are insufficient. The authors should specify how they calculated relative risks, absolute risk differences, and 95% CIs, in addition to the Chi-squared tests and t-tests already described.

Results

- Statistical results for safety data should be reported in greater detail, including relative risks or absolute risk differences, similar to the efficacy outcomes.

- Given the inclusion of bipolar disorder patients, subgroup analyses should be conducted and reported separately for this population. Previous literature on rTMS for bipolar disorder is limited, and such analyses would add value, particularly in mixed cohorts of major depressive disorder and bipolar disorder patients.

Discussion

- The authors state that "the limited number of sessions prevented a more pronounced differentiation between groups." Previous research (e.g., Hsu TW, Neurosci Biobehav Rev, 2024 Jul;162:105704) supports the idea that a greater number of sessions can enhance overall efficacy. The authors are encouraged to elaborate further on this point.

- The recruitment of ~20% bipolar disorder patients warrants additional discussion. A recent meta-analysis (Hsu CW, Neurosci Biobehav Rev, 2024 Jan;156:105483) suggests that high-frequency stimulation of the left DLPFC may not be optimal for bipolar disorder, potentially impacting overall efficacy. The authors should explore this issue in greater depth.

Reviewer #2: This is a multicenter, double blind, randomized (1:1) controlled trial based on a smaller trial which gave the promise of a response advantage of a new intervention over an established procedure in MDD. The study which was modified intent to treat (MITT) was well designed and well planned. Unfortunately it fell short of accrual and the comparative effect expected. The investigators did well describing the limitations of both the previous small study and their own trial as well. The sample size was too small to attempt perhaps some adjusted analysis given the patient characteristics. However, even so, it probably would not have modified the results.

There was mention in the manuscript of a sensitivity study analysis using a per protocol approach with this study which really did not impact the results. However, the protocol included in the supplement was only three pages and lacked much detail, for example, specifically what was the per protocol population. Thus it was difficult to determine the differences between the two strategies.

6. PLOS authors have the option to publish the peer review history of their article (what does this mean? ). If published, this will include your full peer review and any attached files.

**Do you want your identity to be public for this peer review?** For information about this choice, including consent withdrawal, please see our Privacy Policy .

Reviewer #1: No

Reviewer #2: No

- - - - -

---

## [Author Response · Author response to Decision Letter 1]

6 Feb 2025

Please see the file attached for colors.

Dear Editors of PLOS ONE,

We respectfully appeal the decision to reject our manuscript, as we believe it meets the criteria for publication in PLOS ONE. Our study, a randomized controlled trial, presents original findings on the use of MRI-based neuronavigation for rTMS targeting in major depressive episodes. In response to the reviewers’ feedback, we have carefully addressed each of the points raised. We are confident that these revisions have substantially improved the manuscript and that it now aligns with PLOS ONE’s standards for publication.

As requested, we have included the following documents for your review:

1. Appeal Request Form

2. Response to Reviewers

3. Track Changes Version of the Manuscript (and, in addition, the Statistical Analysis Plan)

Thank you for considering our appeal. We look forward to your feedback.

Sincerely,

Professor Florian Naudet

(On behalf of all authors)

Reason for appeal

The reason provided for the rejection was as follows:

"After careful consideration, we have decided that your manuscript does not meet our criteria for publication and must therefore be rejected."

We respectfully disagree with this decision, as we believe our manuscript meets all the PLOS criteria for publication, which can be found here: PLOS ONE Criteria for Publication. Below, we provide a detailed response to each of these criteria, along with comments on how our manuscript aligns with them:

1. The study presents the results of original research.

Our study is a Randomized Controlled Trial (RCT), presenting original findings.

2. Results reported have not been published elsewhere.

The results of this study have not been published elsewhere.

3. Experiments, statistics, and other analyses are performed to a high technical standard and are described in sufficient detail.

In response to the question, “Has the statistical analysis been performed appropriately and rigorously?” the reviewers’ feedback was as follows:

o Reviewer #1: Yes

o Reviewer #2: Yes

4. Conclusions are presented in an appropriate fashion and are supported by the data.

Regarding the question, “Is the manuscript technically sound, and do the data support the conclusions?” the reviewers responded:

o Reviewer #1: Yes

o Reviewer #2: Partly

We would like to note that we can easily address Reviewer #2's concerns and provide further clarifications. However, we do not believe these comments alone justify a rejection, especially given our willingness to respond to feedback.

5. The article is presented in an intelligible fashion and is written in standard English.

In response to the question, “Is the manuscript presented in an intelligible fashion and written in standard English?” the reviewers responded:

o Reviewer #1: Yes

o Reviewer #2: Yes

We are confident that the manuscript meets the language and clarity standards required for publication.

6. The research meets all applicable standards for the ethics of experimentation and research integrity.

Although there was no specific question about ethics, the reviewers raised no concerns regarding this matter. The study was approved by an ethics committee, and informed consent was obtained from all patients whose data are reported. Furthermore, the study adhered to the Good Clinical Practice guidelines, which are the standard for French academic research.

7. The article adheres to appropriate reporting guidelines and community standards for data availability.

In response to the question, “Have the authors made all data underlying the findings in their manuscript fully available?” the reviewers responded:

o Reviewer #1: Yes

o Reviewer #2: Yes

We have ensured that all data underlying our findings are fully available, as required by PLOS Data policies.

Given that our manuscript adheres to all the relevant criteria for publication, we respectfully request a re-evaluation of the decision. Below, we provide more detailed answers to the reviewers’ comments for your consideration.

Point by point response to the reviewer

Reviewer #1: This multicenter, double-blind randomized controlled trial compared MRI-based neuronavigation with the standard 5 cm method for rTMS targeting in major depressive disorder. While the study findings did not support the authors’ hypothesis, the results remain valuable in advancing our understanding of rTMS targeting. To enhance the quality of this manuscript, several major and minor concerns should be addressed.

Major Concerns

- The title refers to "major depressive disorder," and the recruitment criteria cite " major depressive disorder according to DSM-IV criteria in a current depressive episode." However, Table 1 indicates that approximately 60% of participants were diagnosed with persistent depressive disorder and 20% with bipolar disorder. Since persistent depressive disorder and bipolar disorder are distinct from major depressive disorder, the title should be revised accordingly.

We would like to thank the reviewer for this request to clarify our manuscript and agree to revise the term MAJOR DEPRESSIVE DISORDER in MAJOR DEPRESSIVE EPISODE if it is clearer. Indeed, while major depressive disorder is often usen even for bipolar and unipolar depression, the correct term is major depressive episode.

We have revised the title accordingly:

“Repetitive Transcranial Magnetic Stimulation targeted with MRI based neuro-navigation in major depressive episode: a double-blind, multicenter randomized controlled trial”

- The term "persistent depressive disorder" is not part of DSM-IV terminology, which uses "dysthymic disorder." The authors should clarify whether they used DSM-IV or DSM-5 criteria.

We are sorry for the confusion. This is surely a misunderstanding due to an inaccuracy in Table 1 as described in the reviewer’s next comment. Actually all patients had major depressive episode (according DSM-IV) as it was explicitly stated in the text:

“All patients were in- and out- patients, aged from 18 to 65, right-handed, suffering major depressive disorder according to DSM-IV criteria in a current mood depressive episode.”

We confirm that we used the DSM-IV criteria that were in use in France when we initiated the study.

Accordingly, no change has been made to our selection criteria. Please see our response to the next comment for further clarification.

- If participants with persistent depressive disorder were indeed diagnosed solely with dysthymic disorder, this implies that these individuals may not have experienced a major depressive episode. While the recruitment threshold of ≥21 on the Montgomery-Åsberg Depression Rating Scale appears appropriate, the diagnostic criteria and severity thresholds need to align clearly.

We thank the reviewer for this comment. As the reviewer knows, dysthymic disorder and major depressive episode are mutually exclusive. Within DSM-IV. As a reminder we have copied below DSM-IV criteria and outlined in red the fact that “the disturbance is not better accounted for by MDD or MDD in partial remission.”

DSM-IV

Name: Dysthymic Disorder

Class: Mood Disorders

Depressed mood for most of the day, for more days than not, as indicated by subjective account or observation by others, for at least 2 years.

Presence while depressed of two or more of the following:

• Poor appetite or overeating

• Insomnia or hypersomnia

• Low energy or fatigue

• Low self-esteem

• Poor concentration or difficulty making decisions

• Feelings of hopelessness

During the 2 year period of the disturbance, the person has never been without symptoms from the above two criteria for more than 2 months at a time.

The disturbance is not better accounted for by MDD or MDD in partial remission.

There has never been a manic episode, a mixed episode, or a hypomanic episode and the criteria for cyclothymia have never been met.

The disturbance does not occur exclusively during the course of a chronic psychotic disorder.

The disturbance is not due to the direct physiological effects of a substance (e.g., a drug of abuse or a medication) or a general medical condition.

The symptoms cause clinically significant distress or impairment in important areas of functioning.

While we did not include patients with dysthymic disorder in this study, the reviewer was right to point what looks like an inaccuracy in Table 1 likely due to an error when translating it from French to English language. We describle the 3 different categories:

1/ Isolated unipolar depressive episode

2/ Persistent depressive episode

3/ Bipolar episode

Instead of Persistent depressive episode, category 2 should read recurrent unipolar episode.

We have edited the table accordingly.

“Recurrent depressive disorder”

Minor Concerns

Abstract

- The authors report relative risks with 95% confidence intervals (CIs) but do not describe the statistical methods in the Methods section.

We thank the reviewer for this comment. We have edited the abstract accordingly.

“Context: High-frequency (HF) transcranial magnetic stimulation (rTMS) of the left dorsolateral prefrontal cortex (DLPFC) is widely used in Major Depressive Disorder (MDD). Optimization of its efficacy with a neuro-navigation system has been proposed based on a small randomized controlled trial (RCT) supporting a large effect. Method: This evaluator- and patient-blind, multicenter RCT assessed the superiority in terms of efficacy of 10 HF rTMS sessions of the left DLPFC targeted with MRI based neuro-navigation versus similar sessions targeted by the standard 5 cm technique. The study was conducted between January 2013 and April 2017, at 4 hospitals centers in France where both in- and out- patients with MDD were included. Randomization was computer-generated (1:1), with allocation concealment implemented within the e-CRF. The main outcome measure was the percentage of responders 44 days (D44) after the rTMS session. Secondary outcomes were percentage of remitters, Beck Depression Inventory and psychomotor retardation assessed with Salpêtrière retardation rating scale (SRRS) for depression at D14 and D44. The results are presented along with their 95% confidence intervals. Results: 105 patients were randomized and 92 were evaluable with respectively 45 patients in the neuronavigation group and 47 in the standard group. A treatment response was observed for 14 (31.8%) of 44 patients analyzed in the intervention group, and for 16 (35.6%) of 45 patients analyzed in the control group with no statistical difference (relative risk 0.89; 95% confidence interval, [0.50;1.61]). No difference was evidenced for secondary outcomes at D44 whether it concerns remission at D44 (relative risk, 0.82; 95% CI, 0.36 to 1.88), or BDI results (difference in means, 0,01; 95% CI, -3.06 to 3.26), or SRRS results (difference in means, 0.11; 95% CI, -2.42 to 5.02). Similar results were observed at D14. Rates of adverse events were similar in both groups with 23 (47.92 %) and 1 (2.08%) of adverse events and serious adverse events in the neuro-navigation group versus 20 (40.82 %) and 0 (0%) in the standard group. Discussion: This study failed to reproduce previous findings supporting the use of neuro-navigation system to optimize rTMS efficacy. Limitations of this study includes a small sample size and a number of rTMS sessions that may appear substandard in 2020.

Trial registration:

NCT01677078

Funding:

Syneika”

- Adverse event rates are described as "similar," yet no statistical results are reported for this comparison. Statistical tests should be included for consistency.

Statistical tests were included in the text as required to account for the reviewer’s comment, although we think that for rare events such as serious adverse events, results of statistical tests are not informative.

“Rates of adverse events were similar in both groups with 23 (44%) and 1 (2%) of adverse events and serious adverse events in the neuronavigation group versus 20 (38%) and 0 (0%) in the standard group (p-values of 0.55 and 0.49 respectively). Table 3 presents all adverse events.”

Thanks to this comment, we also found an inaccuracy in table 3 with the patient with a SAE also counted in AE section. We have corrected this inaccuracy.

23 (46%)

Please also note that we used for those events Fisher’s exact test and have edited the methods section accordingly.

“The primary outcomes as well as all other categorical efficacy outcomes were compared using Chi2. Safety outcomes were analyzed using Fisher’s exact test.”

Introduction

- The authors aim to "reproduce the findings from Fitzgerald et al." However, their rTMS protocol (20 Hz with 40 pulses per train, 80 trains, inter-train interval of 10 seconds, 110% resting motor threshold, and 10 sessions) differs significantly from that used in Fitzgerald et al. These differences should be explained and acknowledged in the Introduction.

We thank the reviewer for this conceptual point. While we aimed to reproduce the effect identified by Fitzgerald et al. -the superiority of a neuronavigation system-, we did this in a different setting, by using the parameters that were in use in France at this time. We have added this perspective briefly in the introduction.

“We therefore designed this study to reproduce the findings from Fitzgerald et al. (Paul B Fitzgerald et al., 2009) in the French context using a larger sample size and an in-house developed French neuro-navigation system (Syneika; syneika.fr) which takes its originality in the automatic location of the left DLPFC. Thus, the definition of reproducibility includes both the concept of result reproducibility and generalizability, as outlined by Goodman et al. (Goodman et al. 2016). A first objective of this multicenter randomized control trial was to demonstrate the superiority in terms of efficacy over the depressive symptoms of HR-rTMS of the left DLPFC targeted with MRI based neuro-navigation versus the standard localization technique in MDD using the MADRS score.”

REFERENCE

Goodman SN, Fanelli D, Ioannidis JP. What does research reproducibility mean? Sci Transl Med. 2016 Jun 1;8(341):341ps12. doi: 10.1126/scitranslmed.aaf5027. PMID: 27252173.

We have also discussed the differences in the discussion section, following the reviewer’s next comment.

Methods

- The rationale for using 20 Hz rather than the more common 10 Hz protocol should be provided. While 20 Hz is not inherently problematic, justification for its selection is necessary.

We thank the reviewer for this request to clarify the parameters that were chosen. We selected 20-Hz stimulation for this protocol. At the time of writing the study, this choice was based on data from Speer et al. (2000) (Speer et al., 2000), which demonstrated that two weeks of daily 20-Hz rTMS over the left prefrontal cortex at 100% of the motor threshold resulted in a sustained increase in regional cerebral blood flow (rCBF) in the bilateral frontal, limbic, and paralimbic regions implicated in depression. This decision was also supported by multiple studies on the efficacy of 20-Hz rTMS administered over the left prefrontal cortex (Fitzgerald et al., 2003; Garcia-Toro et al., 2001; Padberg et al., 1999). We have edited the text accordingly.

“The rTMS protocol used a frequency of 20 Hz with 40 pulses per train, 80 trains, and an inter-train interval of 10 seconds, resulting in a total of 3,200 pulses per session. The treatment intensity was set at 110% of the patient's resting motor threshold (RMT) as determined by visual observation. The rationale for using 20 Hz was based on data from Speer et al. (2000) (Speer et al., 2000), which demonstrated that two weeks of daily 20-Hz rTMS over the left prefrontal cortex at 100% of the motor threshold resulted in a sustained increase in regional cerebral blood flow in the bilateral frontal, limbic, and paralimbic regions implicated in depression. This decision was also supported by multiple studies on the efficacy of 20-Hz rTMS administered over the left prefrontal cortex (Fitzgerald et al., 2003; Garcia-Toro et al., 2001; Padberg et al., 1999).”

REFERENCES

Fitzgerald, P. B., Brown, T. L., Marston, N. A. U., Daskalakis, Z. J., De Castella, A., & Kulkarni, J. (2003). Transcranial Magnetic Stimulati

---

## [Decision Letter · Decision Letter 1]

24 Mar 2025

PONE-D-24-60061R1Repetitive Transcranial Magnetic Stimulation targeted with MRI based neuro-navigation in major depressive episode: a double-blind, multicenter randomized controlled trialPLOS ONE

Dear Dr. Naudet,

Thank you for submitting your manuscript to PLOS ONE. After careful consideration, we feel that it has merit but does not fully meet PLOS ONE’s publication criteria as it currently stands. Therefore, we invite you to submit a revised version of the manuscript that addresses the points raised during the review process.

We look forward to receiving your revised manuscript.

Kind regards,

Xianwei Che

Academic Editor

PLOS ONE

Journal Requirements:

2. Thank you for providing the following Funding Statement:

"This study was funded by Syneika. The funders had no role in study design, data collection and analysis, decision to publish, or preparation of the manuscript."

We note that one or more of the authors is affiliated with the funding organization, indicating the funder may have had some role in the design, data collection, analysis or preparation of your manuscript for publication; in other words, the funder played an indirect role through the participation of the co-authors.

If the funding organization did not play a role in the study design, data collection and analysis, decision to publish, or preparation of the manuscript and only provided financial support in the form of authors' salaries and/or research materials, please review your statements relating to the author contributions, and ensure you have specifically and accurately indicated the role(s) that these authors had in your study in the Author Contributions section of the online submission form. Please make any necessary amendments directly within this section of the online submission form. Please also update your Funding Statement to include the following statement: “The funder provided support in the form of salaries for authors [insert relevant initials], but did not have any additional role in the study design, data collection and analysis, decision to publish, or preparation of the manuscript. The specific roles of these authors are articulated in the ‘author contributions’ section.”

If the funding organization did have an additional role, please state and explain that role within your Funding Statement.

Please also provide an updated Competing Interests Statement declaring this commercial affiliation along with any other relevant declarations relating to employment, consultancy, patents, products in development, or marketed products, etc.

Within your Competing Interests Statement, please confirm that this commercial affiliation does not alter your adherence to all PLOS ONE policies on sharing data and materials by including the following statement: ""This does not alter our adherence to PLOS ONE policies on sharing data and materials.” (as detailed online in our guide for authors http://journals.plos.org/plosone/s/competing-interests). If this adherence statement is not accurate and there are restrictions on sharing of data and/or materials, please state these. Please note that we cannot proceed with consideration of your article until this information has been declared.

3. In the online submission form, you indicated that [This study involves human participant data that cannot be publicly shared due to French regulatory constraints, as it includes potentially identifying or sensitive patient information. However, datasets will be made available upon request, subject to the non-opposition of study participants. This non-opposition process may incur minimal additional costs. Data requests can be sent to the Data Privacy Officer at Rennes University Hospital at dpo@chu-rennes.fr.].

4. We note that the original protocol that you have uploaded as a Supporting Information file contains an institutional logo. As this logo is likely copyrighted, we ask that you please remove it from this file and upload an updated version upon resubmission.

Additional Editor Comments (if provided):

Reviewers' comments:

Reviewer's Responses to Questions

**Comments to the Author**

1. If the authors have adequately addressed your comments raised in a previous round of review and you feel that this manuscript is now acceptable for publication, you may indicate that here to bypass the “Comments to the Author” section, enter your conflict of interest statement in the “Confidential to Editor” section, and submit your "Accept" recommendation.

Reviewer #1: All comments have been addressed

Reviewer #2: (No Response)

Reviewer #3: All comments have been addressed

2. Is the manuscript technically sound, and do the data support the conclusions?

Reviewer #1: Yes

Reviewer #2: Yes

Reviewer #3: Yes

3. Has the statistical analysis been performed appropriately and rigorously? 

Reviewer #1: Yes

Reviewer #2: Yes

Reviewer #3: Yes

4. Have the authors made all data underlying the findings in their manuscript fully available?

Reviewer #1: Yes

Reviewer #2: Yes

Reviewer #3: Yes

5. Is the manuscript presented in an intelligible fashion and written in standard English?

Reviewer #1: Yes

Reviewer #2: Yes

Reviewer #3: Yes

6. Review Comments to the Author

Reviewer #1: Thank you for revising the manuscript and clarifying the diagnostic terminology. I would like to address two specific issues:

1. Major Depressive Disorder vs. Major Depressive Episode

In your response, you mention that “major depressive disorder is often used even for bipolar and unipolar depression.” While it is sometimes colloquially used in an imprecise manner, strictly speaking, within the DSM classification system, “Major Depressive Disorder (MDD)” should not include episodes that occur as part of bipolar disorder. In DSM-IV, once a patient has experienced a manic or hypomanic episode, the diagnosis shifts to bipolar disorder, rather than MDD. Therefore, using “major depressive disorder” to encompass both unipolar and bipolar depression is not correct from a nosological standpoint. The more accurate term when referring broadly to depressive episodes—whether unipolar or bipolar—is “major depressive episode.”

2. Table 1

You now list three categories: Isolated unipolar depressive episode, Recurrent depressive disorder, and Bipolar disorder. From a diagnostic perspective:

- Isolated unipolar depressive episode typically corresponds to a first or single episode of major depression (i.e., “Major Depressive Disorder, Single Episode” in DSM terminology).

- Recurrent depressive disorder refers to multiple major depressive episodes without a history of mania or hypomania (i.e., “Major Depressive Disorder, Recurrent” in DSM terms).

- Bipolar disorder is used if there has ever been a manic, hypomanic, or mixed episode.

This tripartite classification appears to align more closely with DSM-IV constructs (assuming you distinguish the first unipolar depressive episode from subsequent recurrent unipolar episodes). If your study inclusion criteria specifically required a current major depressive episode, then labeling participants according to these three categories is reasonable, provided you have confirmed that those in the “bipolar disorder” group had a history of mania/hypomania and those in the unipolar groups did not. It would be advisable to ensure that the terms used in Table 1 match official DSM-IV language as closely as possible (e.g., “single-episode major depressive disorder,” “recurrent major depressive disorder,” and “bipolar disorder”).

Reviewer #2: My review is unchanged.

XXXXXXXXXXXXXXXXXXXXXXXXXXXXXXXXXXXXXXXXXXXXXXXXXXXXXXXXXXXXXXXXXXXXXXXXXXXXXXXXXX

XXXXXXXXXXXXXXXXXXXXXXXXXXXXXXXXXXXXXXXXXXXXXXXXXXXXXXXXXXXXXXXXXXXXXXXXXXXXXXXXXX

Reviewer #3: This study compared the efficacy of MRI-based neuronavigation technology and the standard 5 cm technique as targets for high-frequency transcranial magnetic stimulation (rTMS) in the treatment of major depressive disorder (MDD) through a double-blind, multicenter, randomized controlled trial. Although this study did not replicate previous findings supporting the optimization of rTMS efficacy using neuronavigation systems, it still provides valuable clinical evidence for the use of rTMS in treating depression. Overall, the manuscript offers important academic insights, is well-structured, and presents the content in a rigorous manner. While there are some minor issues that need to be addressed, they do not significantly affect the overall quality of the paper. I recommend that the authors make the necessary revisions based on the comments provided, and once these revisions are made, I will be happy to accept the manuscript for publication.

1.In the background section, has a reference been omitted after this sentence?

"In one neuroimaging study of depressed patients, a prefrontal serotonin deficiency at baseline normalized after treatment with rTMS."

2.This paragraph is repetitive in the discussion section:

"Despite these potential differences, we chose not to conduct subgroup analyses, as they carry a risk of alpha risk inflation and cannot be reliably interpreted when not planned a priori and/or when the results are negative, as in our case. Despite these differences, we chose not to conduct subgroup analyses, as they carry a risk of alpha risk inflation and cannot be reliably interpreted when not planned a priori and/or when the results are negative, as in our case."

7. PLOS authors have the option to publish the peer review history of their article (what does this mean? ). If published, this will include your full peer review and any attached files.

**Do you want your identity to be public for this peer review?** For information about this choice, including consent withdrawal, please see our Privacy Policy .

Reviewer #1: No

Reviewer #2: No

Reviewer #3: **Yes: ** Yulei Xie

---

## [Author Response · Author response to Decision Letter 2]

25 Mar 2025

PONE-D-24-60061R1

Repetitive Transcranial Magnetic Stimulation targeted with MRI based neuro-navigation in major depressive episode: a double-blind, multicenter randomized controlled trial

PLOS ONE

Dear Dr. Naudet,

Thank you for submitting your manuscript to PLOS ONE. After careful consideration, we feel that it has merit but does not fully meet PLOS ONE’s publication criteria as it currently stands. Therefore, we invite you to submit a revised version of the manuscript that addresses the points raised during the review process.

We would like to express our gratitude to the editor and reviewers for their meticulous review of our manuscript. We have thoroughly addressed all the points that were raised.

This has been done.

2. Thank you for providing the following Funding Statement:

"This study was funded by Syneika. The funders had no role in study design, data collection and analysis, decision to publish, or preparation of the manuscript."

We note that one or more of the authors is affiliated with the funding organization, indicating the funder may have had some role in the design, data collection, analysis or preparation of your manuscript for publication; in other words, the funder played an indirect role through the participation of the co-authors.

We are surprised by this comment as no co-author is affiliated with the start-up Syneika. Below is the list of authors along with their respective affiliations:

B Millet,a G Harika-Germaneau,b R Maatoug,a F Naudet,c,d JM Reyman,e V Turmel,e JM Batail,f J Soulabaille,f N Jaafari,b D Drapierf

a: Institut du Cerveau et de la Moelle, UMR 7225, CNRS, INSERM, Sorbonne Université et Département de Psychiatrie Adulte, Groupe Hospitalier Pitié-Salpêtrière, Paris, France.

b: CNRS, Université de Poitiers, Université de Tours, CeRCA, Poitiers, France; Unité de Recherche Clinique Pierre Deniker du Centre Hospitalier Henri Laborit, Poitiers, France.

c: Univ Rennes, CHU Rennes, Inserm, EHESP, Irset (Institut de recherche en santé, environnement et travail) - UMR_S 1085, Rennes, France.

d: Institut Universitaire de France (IUF), Paris, France.

e: Univ Rennes, CHU Rennes, Centre d'investigation Clinique (CIC) Inserm 1414, Rennes University Hospital, Rennes, France.

f: Adult Psychiatry Department, Guillaume-Régnier Hospital, Univ Rennes, CHU Rennes, Centre d'investigation Clinique (CIC) Inserm 1414, Rennes, France.

None of the authors are affiliated with Syneika. One author, BM, has stock options in this startup but receives no salary, as detailed in the COI statement. Despite this transparent disclosure, we confirm that "The funder had no role in study design, data collection and analysis, decision to publish, or preparation of the manuscript."

If the funding organization did not play a role in the study design, data collection and analysis, decision to publish, or preparation of the manuscript and only provided financial support in the form of authors' salaries and/or research materials, please review your statements relating to the author contributions, and ensure you have specifically and accurately indicated the role(s) that these authors had in your study in the Author Contributions section of the online submission form. Please make any necessary amendments directly within this section of the online submission form. Please also update your Funding Statement to include the following statement: “The funder provided support in the form of salaries for authors [insert relevant initials], but did not have any additional role in the study design, data collection and analysis, decision to publish, or preparation of the manuscript. The specific roles of these authors are articulated in the ‘author contributions’ section.”

Such a statement would be innacurate has no author got a salary from Sineyka. We made no change. If you think a change is needed, please clarify the request.

If the funding organization did have an additional role, please state and explain that role within your Funding Statement.

The funding organisation had no additional role.

Please also provide an updated Competing Interests Statement declaring this commercial affiliation along with any other relevant declarations relating to employment, consultancy, patents, products in development, or marketed products, etc.

As none of the authors is employed by Syneika, this change was not made. The relationship between Professor Millet and Dr. Nauczyciel (the spouse of Syneika’s CEO), as well as the role of Dr. Nauczyciel, who is not an author, has been clarified. This aims to address any previously unclear potential conflicts of interest in our study, noting that none of the authors are employed by Syneika. Furthermore, we hope this clarification confirms the accuracy of our statement. We are willing to make further changes if necessary. See our answer to the next question.

We have added the following sentence to the COI disclosure: "This does not alter our adherence to PLOS ONE policies on sharing data and materials." See below :

« All authors have completed the ICMJE uniform disclosure form available at http://www.icmje.org/coi_disclosure.pdf, which can be requested from the corresponding author. The authors declare that they have no conflicts of interest, except for Prof. Millet, who holds stocks in Syneika company. Dr Nauczyciel who is syneika CEO spouse worked on Prof. Millet's team and helped with data acquisition without fulfilling authorship criteria. These potential conflict of interest doe not alter our adherence to PLOS ONE policies on sharing data and materials.»

3. In the online submission form, you indicated that [This study involves human participant data that cannot be publicly shared due to French regulatory constraints, as it includes potentially identifying or sensitive patient information. However, datasets will be made available upon request, subject to the non-opposition of study participants. This non-opposition process may incur minimal additional costs. Data requests can be sent to the Data Privacy Officer at Rennes University Hospital at dpo@chu-rennes.fr.].

We have clarified the description and request for an exemption as detailled in the following text. Please note that all 3 reviewers consider that we adhere to PLOS policy and therefore that this exemption seems to be adequate (because public availability would compromise patient privacy). We have also clarfied the generic email adress in order to ensure that data is FAIR with a permanent address at Rennes University Hospital. Here is a detailled statement.

This study involves human participant data that cannot be publicly shared due to French regulatory constraints, as it includes potentially identifying or sensitive patient information (i.e. public availability would compromise patient privacy). In the European context, under GDPR it is not possible to share openly this sensitive data, even when it is pseudonimysed. Furthermore, this is an old study where data sharing was not pre-planned in the documents approved by the IRB. This implies that consent for data sharing was not requested when itw as initiated and this was approved by the IRB. Any data request will need to seek non-opposition from study participants, as practiced in the French context. Therefore, datasets will be made available upon request, subject to the non-opposition of study participants. This non-opposition process may incur minimal additional costs. Data requests can be sent to the Data Privacy Officer at Rennes University Hospital at DRI@chu-rennes.fr.

4. We note that the original protocol that you have uploaded as a Supporting Information file contains an institutional logo. As this logo is likely copyrighted, we ask that you please remove it from this file and upload an updated version upon resubmission.

We have edited document and deleted the logo accordingly.

This has been done.

Done. We see no retracted paper in the list. If you see one, please let us know.

Reviewer #1: Thank you for revising the manuscript and clarifying the diagnostic terminology.

Thank you for this comment.

I would like to address two specific issues:

1. Major Depressive Disorder vs. Major Depressive Episode

In your response, you mention that “major depressive disorder is often used even for bipolar and unipolar depression.” While it is sometimes colloquially used in an imprecise manner, strictly speaking, within the DSM classification system, “Major Depressive Disorder (MDD)” should not include episodes that occur as part of bipolar disorder. In DSM-IV, once a patient has experienced a manic or hypomanic episode, the diagnosis shifts to bipolar disorder, rather than MDD. Therefore, using “major depressive disorder” to encompass both unipolar and bipolar depression is not correct from a nosological standpoint. The more accurate term when referring broadly to depressive episodes—whether unipolar or bipolar—is “major depressive episode.”

We agree with this comment. That’s why we changed the wording to major depressive episode. For clarity purpose, we have again edited a few occurrences that remained in our text.

2. Table 1

You now list three categories: Isolated unipolar depressive episode, Recurrent depressive disorder, and Bipolar disorder. From a diagnostic perspective:

- Isolated unipolar depressive episode typically corresponds to a first or single episode of major depression (i.e., “Major Depressive Disorder, Single Episode” in DSM terminology).

- Recurrent depressive disorder refers to multiple major depressive episodes without a history of mania or hypomania (i.e., “Major Depressive Disorder, Recurrent” in DSM terms).

- Bipolar disorder is used if there has ever been a manic, hypomanic, or mixed episode.

This tripartite classification appears to align more closely with DSM-IV constructs (assuming you distinguish the first unipolar depressive episode from subsequent recurrent unipolar episodes). If your study inclusion criteria specifically required a current major depressive episode, then labeling participants according to these three categories is reasonable, provided you have confirmed that those in the “bipolar disorder” group had a history of mania/hypomania and those in the unipolar groups did not. It would be advisable to ensure that the terms used in Table 1 match official DSM-IV language as closely as possible (e.g., “single-episode major depressive disorder,” “recurrent major depressive disorder,” and “bipolar disorder”).

Many thanks. You are right and we have edited the table accordingly for clarity purpose.

Reviewer #2: My review is unchanged.

XXXXXXXXXXXXXXXXXXXXXXXXXXXXXXXXXXXXXXXXXXXXXXXXXXXXXXXXXXXXXXXXXXXXXXXXXXXXXXXXXX

XXXXXXXXXXXXXXXXXXXXXXXXXXXXXXXXXXXXXXXXXXXXXXXXXXXXXXXXXXXXXXXXXXXXXXXXXXXXXXXXXX

Many thanks, we hope that all comments were addressed. However, these lines of xxxxx are unclear. Please let us know if there was an error.

Reviewer #3: This study compared the efficacy of MRI-based neuronavigation technology and the standard 5 cm technique as targets for high-frequency transcranial magnetic stimulation (rTMS) in the treatment of major depressive disorder (MDD) through a double-blind, multicenter, randomized controlled trial. Although this study did not replicate previous findings supporting the optimization of rTMS efficacy using neuronavigation systems, it still provides valuable clinical evidence for the use of rTMS in treating depression. Overall, the manuscript offers important academic insights, is well-structured, and presents the content in a rigorous manner. While there are some minor issues that need to be addressed, they do not significantly affect the overall quality of the paper. I recommend that the authors make the necessary revisions based on the comments provided, and once these revisions are made, I will be happy to accept the manuscript for publication.

1.In the background section, has a reference been omitted after this sentence?

"In one neuroimaging study of depressed patients, a prefrontal serotonin deficiency at baseline normalized after treatment with rTMS."

Many thanks, we have added the reference for this sentence.

2.This paragraph is repetitive in the discussion section:

"Despite these potential differences, we chose not to conduct subgroup analyses, as they carry a risk of alpha risk inflation and cannot be reliably interpreted when not planned a priori and/or when the results are negative, as in our case. Despite these differences, we chose not to conduct subgroup analyses, as they carry a risk of alpha risk inflation and cannot be reliably interpreted when not planned a priori and/or when the results are negative, as in our case."

Many thanks, this was corrected.

---

## [Editor Report · Decision Letter 2]

15 Apr 2025

Repetitive Transcranial Magnetic Stimulation targeted with MRI based neuro-navigation in major depressive episode: a double-blind, multicenter randomized controlled trial

PONE-D-24-60061R2

Dear Dr. Naudet,

We’re pleased to inform you that your manuscript has been judged scientifically suitable for publication and will be formally accepted for publication once it meets all outstanding technical requirements.

Kind regards,

Xianwei Che

Academic Editor

PLOS ONE
---

## [Editor Report · Acceptance letter]

PONE-D-24-60061R2

PLOS ONE

Dear Dr. Naudet,

I'm pleased to inform you that your manuscript has been deemed suitable for publication in PLOS ONE. Congratulations! Your manuscript is now being handed over to our production team.

Kind regards,

on behalf of

Dr. PLOS Manuscript Reassignment

Staff Editor

PLOS ONE